# MDBench: A Synthetic Multi-Document Reasoning Benchmark Generated with Knowledge Guidance

## Abstract

Natural language processing evaluation has made significant progress, largely driven by the proliferation of powerful large language models (LLMs). New evaluation benchmarks are of increasing priority as the reasoning capabilities of LLMs are expanding at a rapid pace. In particular, while *multi-document* (MD) reasoning is an area of extreme relevance given LLM capabilities in handling longer-context inputs, few benchmarks exist to rigorously examine model behavior in this setting. Moreover, the multi-document setting is historically challenging for benchmark creation due to the expensive cost of annotating long inputs.

In this work, we introduce **MDBench**, a new dataset for evaluating LLMs on the task of multi-document reasoning. Notably, MDBench is created through a novel synthetic generation process, allowing us to *controllably and efficiently generate challenging document sets* and the corresponding question-answer (QA) examples. Our novel technique operates on condensed structured seed knowledge, modifying it through LLM-assisted edits to induce MD-specific reasoning challenges. We then convert this structured knowledge into a natural text surface form, generating a document set and corresponding QA example. We analyze the behavior of popular LLMs and prompting techniques, finding that MDBench poses significant challenges for all methods, even with relatively short document sets. We also see our knowledge-guided generation technique (1) allows us to readily perform targeted analysis of MD-specific reasoning capabilities and (2) can be adapted quickly to account for new challenges and future modeling improvements.

## 1 Introduction

The rapid advancements in natural language processing (NLP) have been largely driven by the development and deployment of large language models (LLMs). These models have showcased remarkable improvements in various tasks, including understanding, generating, and reasoning over text. However, despite these advancements, evaluation frameworks for NLP systems have struggled to keep pace (Chang et al., 2024), notably for tasks involving reasoning over multiple documents (Mavi et al., 2024).

Multi-document (MD) reasoning involves synthesizing and inferring information across multiple diverse texts (Caciularu et al., 2021), posing unique challenges not addressed by traditional single-document benchmarks. While LLMs are increasingly capable of handling longer-context multi-document inputs, there is a scarcity of benchmarks that rigorously examine the specific reasoning characteristics that are prominent in this setting. In addition, many existing benchmarks consist of static, hand-crafted datasets, which are labor-intensive to produce. These datasets are often susceptible to data contamination (Xu et al., 2024) over time, e.g., LLMs are exposed to public benchmarks during training. This can compromise the integrity of the evaluation.

In this work, we address these limitations with **MDBench**, a benchmark using a novel generation technique for multi-document reasoning evaluation. Our benchmark is generated through a synthetic process that leverages structured knowledge as seed information. This process uses a strong LLM (GPT-4o) to augment structured knowledge by injecting complexities that require advanced reasoning skills, then generates text documents from the augmented knowledge.

Our benchmark generation pipeline begins with a structured knowledge source serving as the seed information. Each knowledge entry (i.e., row of the table) encapsulates distinct knowledge that forms the basis of a document in the generated set. We follow a three-step augmentation process to source knowledge, augment knowledge, and generate document sets with multi-document reasoning challenges:

1. **Source Seed Knowledge:** We collect tabular data where each row contains information that will contribute to a generated document.

2. **Augment Knowledge:** Using a powerful LLM, we edit the structured knowledge to inject challenging reasoning dependencies and enrich the context for document creation. By treating rows as proxies for documents, we model cross-document dependencies through cross-row knowledge interactions. In this step, we also generate question-answer pairs that utilize the introduced reasoning dependencies.

3. **Generate Natural Text:** We map the augmented knowledge into natural text by generating a corresponding multi-document set from the augmented table. This process allows us to systematically inject critical reasoning challenges while producing examples that are realistic and fluent.

We produce a substantial number of multi-document QA examples using this pipeline (300 human-validated, and 700 more automatically-validated for quality) and evaluate the performance of models from several prominent LLM families including GPT, Claude, Gemini, and Llama. We find that:

- MDBench poses a strong challenge, even for state-of-the-art methods, with the best ones achieving ∼59% performance on this MD reasoning task.

- Frontier models such as GPT-4o and Claude Sonnet significantly outperform smaller LLMs across different prompting methods. This highlights the importance of model capacity and sophistication in handling complex multi-document reasoning tasks.

- When comparing performance on document reasoning versus tabular reasoning (i.e., structured format pre-document generation), we find that strong models are mostly performant in both settings. However, smaller models struggle more in the long-form document setting. This suggests that *multi-document reasoning is influenced by both the fundamental reasoning complexity, and also from the nuances of the surface form.*

- Prompting techniques such as Chain-of-Thought (Wei et al., 2022) can improve performance across strong models. However, they are insufficient to significantly enhance the performance of weaker models like Llama3-7B and GPT-3.5. This indicates that while prompting strategies can aid reasoning, *underlying model capabilities remain a limiting factor for this task, which makes MDBench suitable for future, advanced model evaluation.*

## 2 RELATED WORK

Evaluating the capabilities of LLMs is a critical aspect of NLP research. As LLMs continue to improve rapidly, existing evaluation frameworks often lag behind, particularly in assessing complex reasoning abilities such as multi-document (MD) reasoning. As LLMs rapidly increase in reasoning capacity, there is a pressing need to develop evaluation methods that can capture these higher-order reasoning skills.

**Multi-Document Reasoning** MD reasoning involves synthesizing and inferring information across multiple texts. Existing work in this area includes datasets targeting specific phenomena such as temporal reasoning (Xiong et al., 2024; Wan, 2007), summarization (Xiao et al., 2021; Peper et al., 2023; Lior et al., 2024), multi-hop question answering (Yang et al., 2018; Qi et al., 2021; Trivedi et al., 2022) and ambiguous entity resolution (Lee et al., 2024). Notably, many of these MD datasets are publicly-sourced and often reliant on significant human effort to curate For example, Zhu et al. (2024) introduce FanOutQA, a recent multi-hop, multi-document question answering dataset, which targeted decomposable QA examples sourced from public Wikipedia knowledge and relied on thousands of manual annotations. Our work seeks to use knowledge-controlled generation to offer a scalable alternative for producing nuanced and unseen multi-document reasoning examples.

**Tabular Reasoning with LLMs**    LLMs have demonstrated strong performance in tasks involving structured knowledge, such as tabular data or knowledge bases (Lu et al., 2024; Li et al., 2023a). Recent studies have observed success in applying LLMs to table reasoning, manipulation, and augmentation (Lu et al., 2024; Li et al., 2023a). While there are limitations in LLM pre-training which can lead to formatting sensitivities and limitations with handling large tables, Nahid & Rafiei (2024) find improved performance by decomposing the tabular knowledge into a digestible size. Similarly, leveraging tabular knowledge within reasoning chains allows for compact and effective representation of complex problems, as explored in the Chain-of-Tables framework (Wang et al., 2024). These insights highlight the potential of using condensed knowledge as a foundation for generating challenging reasoning tasks.

**LLM-Supported Synthetic Benchmark Creation**    To address the need for more dynamic evaluation datasets, LLM-powered synthetic benchmark creation has gained significant traction (Long et al., 2024; Liu et al., 2024; Li et al., 2023b), particularly as there is growing concern of benchmark data contamination Xu et al. (2024) Some work has been done in the multi-document setting, although automation is largely used for extending existing annotated multi-document benchmarks to more complex tasks (Schnitzler et al., 2024). While not directly modeling multi-document tasks, Sprague et al. (2023) explore synthetic generation in the related multi-step reasoning setting, using a neurosymbolic generation algorithm which maps synthetic structure into natural text examples. Our method seeks to build off related work in synthetic generation to address efficient multi-document benchmark creation.

## 3    MDBench Generation Pipeline

In this section, we motivate and overview the generation process, and provide details on the components and steps taken to produce the MDBench evaluation benchmark.

### 3.1    Benchmark Generation Goals

- **Contain Novel and Unseen Text**: We aim to produce examples that are not merely scraped from public datasets but rather contain newly-generated content. This ensures that models are tested on scenarios they have not encountered during training, avoiding overfitting to pre-existing benchmarks.

- **Contains Cross-Document Knowledge Dependencies**: A key focus is to produce examples that require reasoning across multiple documents. We design our benchmark to have intentional cross-document dependencies, making them particularly challenging, testing multi-document reasoning capabilities.

- **Grounded in Real-World Scenarios**: Even though the examples are synthetically generated, they should ideally remain grounded in real-world concepts and situations. This ensures that the reasoning challenges presented are realistic and relevant to practical NLP applications.

- **Counterfactual Alterations**: To further mitigate data contamination and leakage risks from public sources, we incorporate slight counterfactual or fictional twists on real-world scenarios. This allows for a fresh take on familiar domains while maintaining the integrity of the benchmark.

- **Scalability and Control**: Our approach is designed to offer control during benchmark generation. We allow one to specify seed information such as domain and behavior types, and can control the complexity and nature of the reasoning tasks present in the benchmark.

### 3.2    Pipeline Overview

Our benchmark generation pipeline begins with structured knowledge sourced from tabular data, which serves as the seed for the augmentation process. This structured knowledge is systematically enriched and refined through a strong LLM to inject reasoning dependencies that challenge models to infer information across multiple documents. Figure 1 overviews the pipeline.

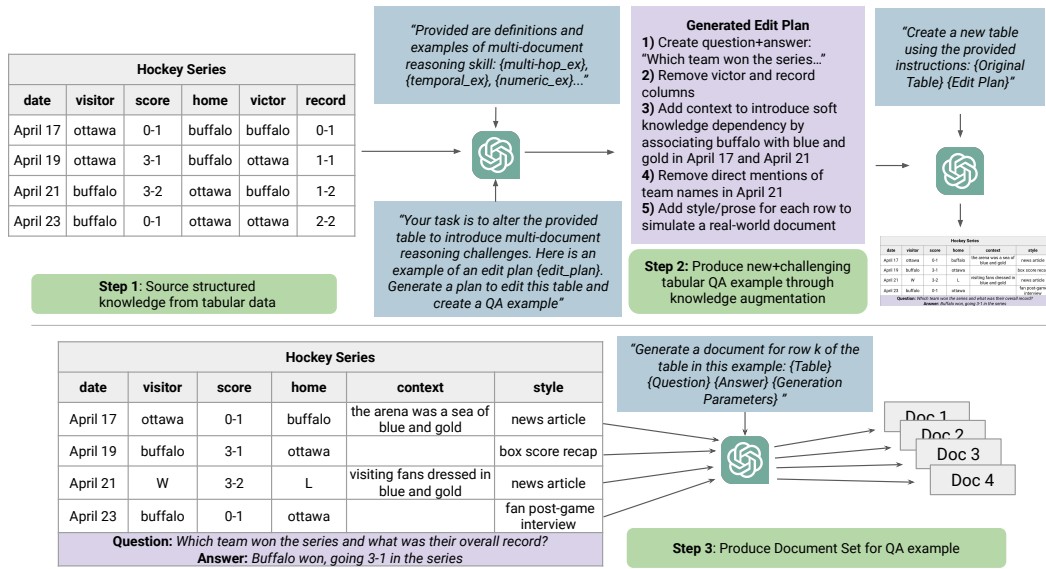

Figure 1: MDBench generation pipeline overview. We source structured knowledge, then use in-context multi-document reasoning demonstrations to intentionally modify the existing knowledge with challenging dependencies. We then map this seed knowledge into document form to produce the multi-document QA example.

**Step 1: Obtaining Seed Knowledge** We start with the intuition that compressed structured knowledge provides an effective foundation for multi-document reasoning. Several valid sources of this exist, such as knowledge bases, tabular information, or even by performing information extraction to consolidate data from existing documents and text corpora. For the MDBench benchmark, we utilize the TabFact (Chen et al., 2020) dataset, which comprises 16,000 tables sourced from Wikipedia. Our motivation for exploring this dataset is threefold: (1) TabFact tables provide a reliable and curated source of seed knowledge (2) the data spans a wide range of domains, including news, sports, media, and technology, and (3) has an emphasis on human-readability both in scale and content. This structured knowledge serves as the starting point for our knowledge augmentation process, which significantly transforms the raw data into more challenging and complex reasoning tasks. We heuristically filter the dataset to select tables that are rich in content yet manageable in size, choosing those with 5 to 15 rows and 3 to 8 columns.

**Step 2: Knowledge Augmentation** An important component of our technique is the knowledge augmentation step. This step modifies information, applying operations that inject complex knowledge dependencies and reasoning challenges. Figure 1 overviews our pipeline, while full detailed examples of the knowledge augmentation prompts are provided in Appendix A.

- **Multi-document Reasoning Demonstrations** Prior to altering the existing information we first demonstrate relevant skills for multi-document reasoning. Each skill is demonstrated in both 'simple' and 'challenging' forms. The demonstrations include examples, along with explanations and rationales for solving them. For the purpose of this benchmark, we define and emphasize five reasoning components which are particularly relevant in the multi-document setting. For each skill we demonstrate both a simple and more complex example, each highlighting the relevant reasoning. We describe these skills in Table 2.

- **Knowledge Augmentation Demonstrations** In addition to demonstrating relevant reasoning skills, we next provide *knowledge edit demonstrations*. These demonstrations illustrate plans for how simple tables can be enhanced to form nuanced QA examples. Each demonstration consists of an initial table, a series of edits, and a resultant augmented table and QA annotation. When performing knowledge augmentation, we provide one demonstration from a small set of high-quality curated examples.

**Baseline Example:** Which country had the most showings and how many was this in total?

| date | territory | showings |
|---|---|---|
| october 20, 2006 | turkey | 200 |
| october 20, 2006 | belgium | 600 |

**Answer:** Belgium had the most with 600 showings.
**Answer rationale:** Turkey had 200 showings and Belgium had 600. 600 > 200, therefore Turkey had the most showings.
**Commentary:** This is a simple reasoning process as it requires a simple comparison of two values with no additional reasoning required.

**Harder Example:** Which country had the most showings and how many was this in total?

| date | territory | showings |
|---|---|---|
| october 20, 2006 | turkey | 200 |
| october 20, 2006 | belgium | 600 |
| october 25, 2006 | turkey | 500 |

**Answer**: Turkey had the most with 700 showings.
**Answer Rationale:** Turkey had showings on two different days, so the total is 200+500=700 showings. 700 > Belgium's 600, therefore Turkey had the most.
**Commentary:** By adding a new row with complementary information, we necessitate an additional reasoning hop to correctly answer the question. Note that this table was edited specifically such that the answer (Turkey) is flipped from the original answer (Belgium) in the simple example.

Figure 2: Example Skill Description – Multi-hop Reasoning. During knowledge augmentations, we demonstrate the multi-document skills relevant to the document sets.

.

| | Min | Mean (std) | Max |
|---|---|---|---|
| # Docs / Rows | 5 | 8.79 (2.5) | 17 |
| # Table Columns | 3 | 5.42 (1.2) | 9 |
| Token Length (Tabular Format) | 121 | 255.98 (81.6) | 554 |
| Token Length (Doc. Format) | 1048 | 2317.81 (754.1) | 6210 |
| Avg. Document Token Length | 177 | 264.02 (37.6) | 388 |

Table 1: MDBench benchmark statistics. Each row in the tabular representation ultimately corresponds to a document within the multi-document example. We see a roughly 9x increase in surface form length when mapping the structured knowledge to natural text document format.

Through these two steps, we modify the tabular knowledge to form a more nuanced QA example with cross-row knowledge dependencies.

**Step 3: Document Set Generation** Once the tabular knowledge has been augmented, we map this information into natural language text; each row in the table is used to generate a document, with the augmented knowledge ensuring that reasoning across documents (rows) is required to solve the accompanying QA task. We independently generate each document, the generation prompt parameterized by the following components: (1) the augmented table and title, (2) the column names and (3) a specific row of content within the table indicated for generation. Iterating this process over all $n$ rows in the table, we generate an $n$-document set. This approach of knowledge-grounded generation ensures the generated document set maintains logical coherence while presenting unique cross-document reasoning challenges.

### 3.3 MDBench Benchmark Generation Details

We use GPT-4o as the backbone of the pipeline, for both table augmentation and document generation. We note that quality control is a crucial process for synthetic data generation (Long et al., 2024), and we use automated validation steps in both generation steps to mitigate compounding errors within the pipeline. We generate and hand-verify 300 produced examples, and also produce 700 machine-validated examples for community use. Details of the automated validation prompts used in the generation process are outlined in Appendix C. Table 1 outlines the statistics of the generated benchmark.

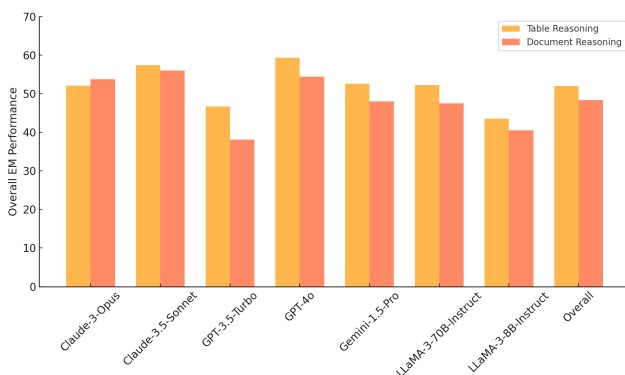

Figure 3: Overall performance of models on MDBench. Table reasoning is when evaluated with the intermediate table QA examples. Document Reasoning refers to the performance on the final task of multi-document reasoning.

| Reasoning Type | Description |
|---|---|
| Multi-hop Reasoning | Solving problems requiring multiple steps to arrive at the solution. |
| Numeric Reasoning | Handling numeric values and performing numerical operations. |
| Temporal Reasoning | Handle temporal information and temporal dependencies. |
| Knowledge Aggregation | Aligning, comparing and/or contrasting knowledge that may be present. |
| Soft Reasoning | Reasoning abductively and making informed decisions in cases where some uncertainty or fuzziness may be present, such as cross-document entity linking. |

Table 2: Reasoning skills overview. For our benchmark, we focus on five goals which are especially relevant for the multi-document setting. We provide demonstrations of these reasoning types to inspire relevant knowledge edits during the generation process.

## 4 EXPERIMENTAL SETUP

To assess the challenges of MDBench, we test the performance of many popular LLMs in combination with conventional prompting setups. Concretely, we test open-source LLMs with Meta's Llama-3 (Dubey et al., 2024), using the 8B-Instruct and 70B-Instruct variants. For API-based proprietary models, we use models from the popular Anthropic Claude, OpenAI GPT, and Google Gemini model familes, which represent the state-of-the-art in LLM performance. For Claude, we use Claude-3-Opus-20240229 and Claude-3.5-Sonnet-20240620[1]. For GPT we use GPT-3.5-turbo-16k-0613[2] (Ouyang et al., 2022) and GPT-4o-2024-08-06[3]. For Gemini, we use Gemini-1.5-Pro-0514 (Team et al., 2024).

We explore both *zero-shot* and *one-shot* QA prompting scenarios, noting that when prompting in the one-shot case we use a single representative demonstration across models for consistency. We use a conventional question-answering prompt, and also further instruct the models to 'think step by step' to additionally produce *Chain-of-Thought* (CoT) rationales. Examples of these prompt formats are provided in Appendix B. To evaluate on the QA task, we use GPT-4o as a reference-based scorer, first parsing the final answer from each output, then comparing the similarity of the predicted answer with the ground-truth answer (conditioned on the original question). We calculate both an *exact match* score as well as an *accuracy* score, where the scorer can assign partial correctness credit on a 1-10 scale.

---

[1]https://www.anthropic.com/claude

[2]https://platform.openai.com/docs/models/gpt-3-5

[3]https://platform.openai.com/docs/models/gpt-4o

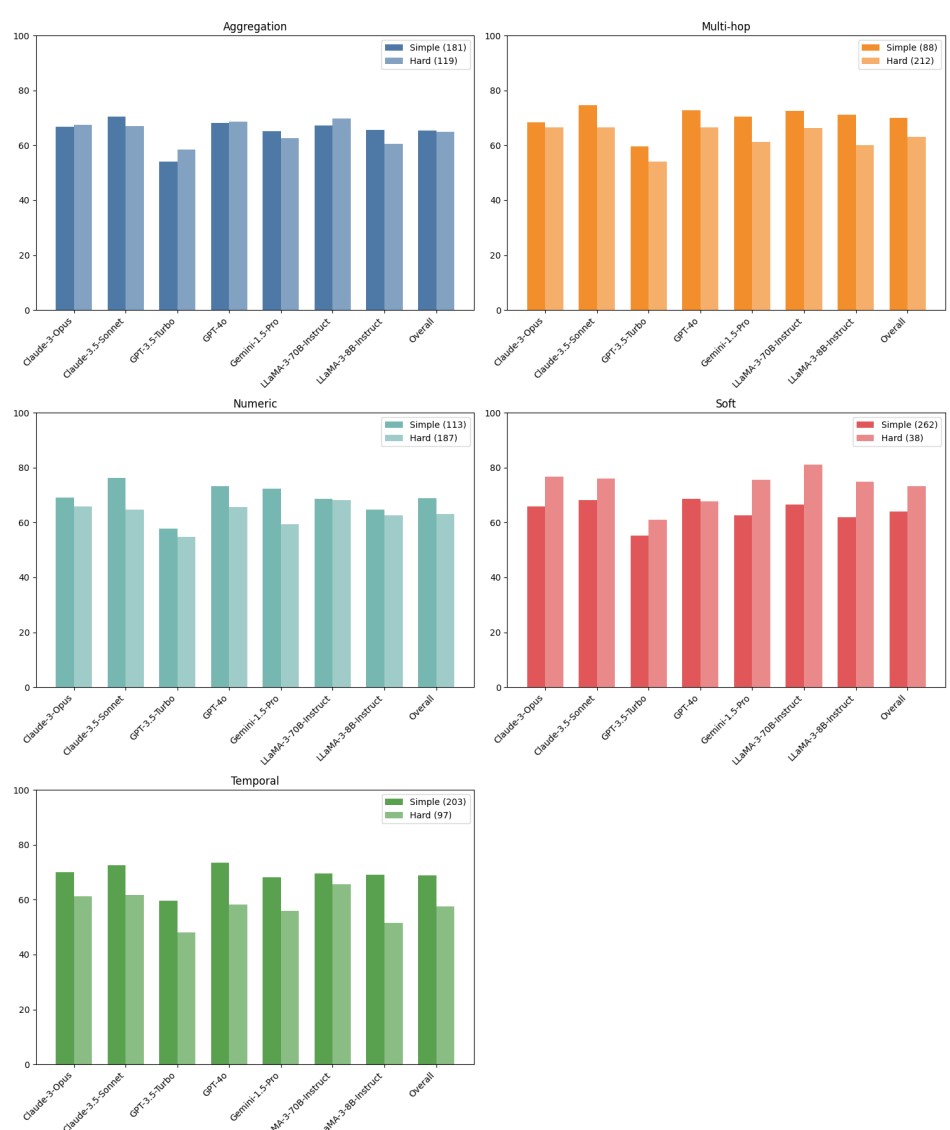

Figure 4: Characteristic-level performance breakdown. We report each model's overall accuracy on each of the bins.

## 5 RESULTS + ANALYSIS

**Overall Findings** Figure 3 and Table 3 overview the performance on our new multi-document reasoning benchmark. MDBench poses a strong challenge, even for state-of-the-art methods, with the best methods achieving ∼59% exact-match performance. Claude-3.5-Sonnet performs best overall on the document reasoning task, with 54.4% overall performance. Sonnet performs strong on all splits. Notably, we see mixed benefits to Chain-of-Thought for weaker models, where in comparison, Chain-of-Thought is usually beneficial for larger models such as Sonnet and GPT-4o, although we observe that most models generally produce reasoning chains even without explicit CoT prompting. Of the large API-based frontier models, we see Gemini-1.5-Pro struggles the most, although it performs relatively well when evaluating on overall accuracy (where partial credit is assigned during scoring). Notably, Llama3-70B performs strongly, outperforming GPT-3.5 in several cases.

**Document vs. Tabular Reasoning** To ascertain the impact of surface form on the reasoning task, we compare the performance of models on the full multi-document version of the benchmark versus

| Model | Zero-shot | Zero-shot CoT | One-shot | One-shot CoT | Overall |
|---|---|---|---|---|---|
| Claude-3-Opus | 52.9 | 51.6 | **58.8** | 51.6 | 53.8 |
| Claude-3.5-Sonnet | 54.9 | **56.9** | 56.2 | **56.2** | **56.0** |
| GPT-3.5-Turbo | 44.4 | 37.3 | 38.6 | 32.0 | 38.1 |
| GPT-4o | **56.9** | **56.9** | 51.0 | 52.9 | 54.4 |
| Gemini-1.5-Pro | 49.0 | 45.8 | 48.4 | 49.0 | 48.0 |
| LLaMA-3-70B-Instruct | 52.6 | 45.1 | 51.9 | 40.5 | 47.5 |
| LLaMA-3-8B-Instruct | 46.4 | 39.2 | 41.8 | 34.6 | 40.5 |

| Model | Zero-shot | Zero-shot CoT | One-shot | One-shot CoT | Overall |
|---|---|---|---|---|---|
| Claude-3-Opus | 67.1 | 64.5 | 68.5 | 64.7 | 66.2 |
| Claude-3.5-Sonnet | **69.1** | **68.8** | **70.3** | **68.4** | **69.2** |
| GPT-3.5-Turbo | 55.9 | 53.9 | 52.2 | 49.8 | 53.0 |
| GPT-4o | 68.5 | 68.0 | 64.7 | 67.5 | 67.2 |
| Gemini-1.5-Pro | 64.1 | 63.3 | 67.3 | 63.7 | 64.6 |
| LLaMA-3-70B-Instruct | 66.3 | 58.4 | 66.4 | 58.4 | 62.4 |
| LLaMA-3-8B-Instruct | 63.5 | 58.2 | 60.4 | 53.3 | 58.8 |

Table 3: Document Reasoning Overall Results. We report exact-match (top) and accuracy (bottom) results on the MDBench multi-document examples.

| Model | Zero-shot | Zero-shot CoT | One-shot | One-shot CoT | Overall |
|---|---|---|---|---|---|
| Claude-3-Opus | 51.0 | 50.3 | 54.2 | 52.9 | 52.1 |
| Claude-3.5-Sonnet | **59.5** | 57.5 | 55.6 | 56.9 | 57.4 |
| GPT-3.5-Turbo | 47.1 | 43.8 | 45.8 | 50.3 | 46.7 |
| GPT-4o | 58.8 | **58.2** | **60.8** | **59.5** | **59.3** |
| Gemini-1.5-Pro | 51.0 | 48.4 | 53.6 | 57.5 | 52.6 |
| LLaMA-3-70B-Instruct | 52.9 | 52.6 | 52.6 | 51.0 | 52.3 |
| LLaMA-3-8B-Instruct | 43.1 | 46.4 | 43.8 | 40.5 | 43.5 |

| Model | Zero-shot | Zero-shot CoT | One-shot | One-shot CoT | Overall |
|---|---|---|---|---|---|
| Claude-3-Opus | 68.3 | 65.0 | 70.3 | 63.5 | 66.8 |
| Claude-3.5-Sonnet | **70.7** | 70.8 | 70.3 | 69.5 | 70.3 |
| GPT-3.5-Turbo | 62.9 | 57.4 | 60.1 | 62.9 | 60.8 |
| GPT-4o | 70.6 | **71.2** | **71.2** | **75.9** | **72.2** |
| Gemini-1.5-Pro | 67.8 | 63.2 | 68.1 | 70.7 | 67.5 |
| LLaMA-3-70B-Instruct | 66.3 | 65.3 | 66.1 | 63.9 | 65.4 |
| LLaMA-3-8B-Instruct | 58.8 | 62.1 | 58.8 | 54.4 | 58.5 z |

Table 4: Table Reasoning Overall Results. We report exact-match (top) and accuracy (bottom) when applying models to the augmented *tabular format* QA examples (as opposed to documents).

the table version (i.e., stopping after step 2 in our pipeline). Table 4 overviews the table-reasoning results, and the comparison of overall results can be seen in Figure 3. We find that performance is generally higher on the condensed tabular format of the dataset. For example, this difference is quite notable for GPT-3.5-Turbo, with a drop from 46.7% to 38.1% EM performance for tabular versus document reasoning. Overall, Sonnet has the highest overall document-reasoning performance, and GPT-4o has the highest table-reasoning performance.

**Characteristic Breakdown** We additionally evaluate the performance as a function of the example difficulty. To do this, we prompt GPT-4o to generate characteristic-level difficulty scores for each example. We use the same five characteristics as demonstrated in the generation process, and prompt the model with these definitions. Rather than generating absolute scores, we instead approximate difficulty by prompting GPT-4o to perform comparative ranking with two other randomly sampled examples for each characteristic. We aggregate these relative rankings over the entire dataset to form two difficulty bins per characteristic, as overviewed in Figure 4.

We see mostly consistent trends across characteristics, with temporal reasoning posing the starkest dropoff between the simple and hard bins. Interestingly, we see soft reasoning is impacted inversely, with performance increasing on the split of examples ranked to have harder soft-reasoning components. While some of this may be due to small sample size for for the hard bin (only 38 of 300 examples), we suspect there is an inverse relationship between soft reasoning and more 'explicit' characteristics such as numeric and temporal. For example, a table/example well-suited for temporal reasoning may naturally contain less 'soft' information requirements. Conversely, an example with significant soft reasoning requirements likely contains fewer hard reasoning requirements.

## 6 CONCLUSION

In this work, we present MDBench, a novel benchmark designed to evaluate large language models on multi-document reasoning tasks. By leveraging structured seed knowledge and augmenting it with nuanced reasoning dependencies, MDBench enables the systematic development of challenging, multi-document QA examples and addresses key challenges in traditional benchmark creation, including issues related to data contamination and the difficulty of efficiently generating diverse reasoning examples. Our work introduces a new method for probing complex cross-document reasoning, paving the way for more rigorous evaluation of models' abilities to handle real-world, multi-source information, and advancing the development of LLMs capable of deeper, contextually aware reasoning.

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

## A   MULTI-DOCUMENT REASONING SKILLS DEMONSTRATIONS

Figures 6, 8, 7, 9, 10 overview the five reasoning skills we demonstrate during the creation of MDBench. Figure 5 demonstrates an edit plan provided to inspire the table augmentation.

## B   MODEL EVALUATION PROMPTS

---

**Simple QA Prompt**

"You will be presented with a question and a context. You should answer the question based on the context. The last thing you generate should be ANSWER:[your answer here]"

---

**Chain-of-thought QA Prompt**

"You will be presented with a question and a context. You should answer the question based on the context. Explain your reasoning step by step before you answer. The last thing you generate should be ANSWER:[your answer here]"

---

| | Aggregation | | Multi-hop | | Numeric | | Soft | | Temporal | |
|---|---|---|---|---|---|---|---|---|---|---|
| Difficulty Level | E | H | E | H | E | H | E | H | E | H |
| Support | 181 | 119 | 88 | 212 | 113 | 187 | 262 | 38 | 203 | 97 |
| Claude-3-Opus | 66.9 | 67.5 | 68.5 | 66.5 | 69.1 | 65.9 | 65.9 | 76.7 | 70.0 | 61.2 |
| Claude-3.5-Sonnet | 70.6 | 67.0 | 74.7 | 66.6 | 76.3 | 64.8 | 68.1 | 76.1 | 72.6 | 61.8 |
| GPT-3.5-Turbo | 54.0 | 58.6 | 59.6 | 54.2 | 57.9 | 54.7 | 55.2 | 61.1 | 59.6 | 48.2 |
| GPT-4o | 68.3 | 68.7 | 72.8 | 66.6 | 73.2 | 65.7 | 68.6 | 67.8 | 73.5 | 58.2 |
| Gemini-1.5-Pro | 65.1 | 62.7 | 70.6 | 61.2 | 72.3 | 59.3 | 62.6 | 75.6 | 68.1 | 56.0 |
| LLaMA-3-70B-Instruct | 67.3 | 69.7 | 72.6 | 66.4 | 68.6 | 68.1 | 66.6 | 81.1 | 69.6 | 65.6 |
| LLaMA-3-8B-Instruct | 65.6 | 60.5 | 71.3 | 60.0 | 64.7 | 62.7 | 61.9 | 75.0 | 69.2 | 51.6 |
| Overall | 65.4 | 64.9 | 70.0 | 63.1 | 68.9 | 63.0 | 64.1 | 73.3 | 68.9 | 57.5 |

Table 5: Characteristic-level Performance Breakdown. We report overall accuracy.

## C  MDBENCH PIPELINE VALIDITY PROMPTS

We use the following prompts during the knowledge augmentation step to validate the edit plan execution and resultant QA example. Prompt 1 works through the generated problem (leveraging the full knowledge augmentation history) and attempts to rationalize the QA example. Then, prompt 2 evaluates whether this rationalization from Prompt 1 is valid and generates a 0-5 validity scalar.

---

**Validity Prompt 1**

Original Table Name: {table_title}
Original Table: {original_table}
Table Edits Applied: {edits_applied}
Resultant Table: {generated_table}
Resultant Question: {generated_question}
Resultant Answer: {generated_answer}

Prompt: I have provided an original table, and then an updated version (using the provided knowledge edits) which resulted in an augmented table with a corresponding new question and answer. Use this context and think step by step to come up with a solution rationale that provides a justification for the answer. Note that the original table + edits are provided mostly for added reference. Output the rationale as a string.

---

**Validity Prompt 2**

How consistent/valid is this reasoning in the following process for generating an example from a table? Score the validity and consistency of the resultant table+question+answer on a scale of 0-5. I want to be able to identify and ignore examples with low scores that I shouldn't include in my dataset. Output as a json with 'score' and 'explanation' fields. Here is the example: {prompt_1_output}

---

## D  CHARACTERISTIC BREAKDOWN

Table 5 overviews the overall model performance when binning examples by difficulty for each of the five considered characteristics.

**Original Table | Table Summary:** Movie Sales by Country

| date | territory | screens | rank | gross ($) |
|---|---|---|---|---|
| october 20 , 2006 | turkey | 378 | 1 | 146268 |
| october 25 , 2006 | belgium | 6 | 19 | 38916 |
| october 25 , 2006 | germany | 52 | 12 | 133228 |
| october 26 , 2006 | austria | 4 | 13 | 41780 |
| october 26 , 2006 | netherlands | 17 | 14 | 53749 |
| october 27 , 2006 | united kingdom | 4 | 24 | 34704 |

**Edit 1:** Come up with a interesting question about this table. The question MUST have a concise verifiable answer. The question should go hand in hand with ensuring the augmentation introduces complex cross-row dependencies, as this will be used to create corresponding multi-document examples (one document per row). Make sure that the question + new table can only be answered if the model reasons correctly over documents.
**Example:** "Rank the movie's sales by country." -- requires reasoning/comparing over the different rows in the document. Note: we will edit the table further to make this even more challenging.

**Edit 2:** Remove extraneous columns to avoid overspecification in the resultant documents
**Example:** Remove the screens and rank columns since they're not relevant

**Edit 3:** Round some of the numeric values to eventually make the information more realistic in the articles
**Example:** Round the gross sales numbers to thousands

**Edit 4:** Add 'multi-hop' information, or additional rows that necessitate synthesizing information across documents
**Example:** Add an October 26th entry for Germany for $195k (now there are two rows for Germany) -- These need to be added into order to calculate the Germany sales.

**Edit 5:** Add secondary / peripheral / fictional information to contextualize/personalize the documents.
**Example:** Add a "context" column with some additional guidance to guide the document generation. This should include instructions on document length + writing style as well as superfluous content that might naturally occur in a document of this type. Also add a fictionalized film name (Nightmares of Glory).

**Edit 6:** Introduce cross-document dependencies by obfuscating some linked information. The dependencies must be utilized within the question answering process.
**Example:** The Germany Oct. 26th entry was modified. The country information was obfuscated, but the daily revenue was defined in terms of the prior day, allowing the model to refer back to the Oct. 25 row.

**Resultant Augmented Table**

| date | country | daily revenue ($) | film | context |
|---|---|---|---|---|
| october 20 , 2006 | turkey | 146200 | Nightmares of Glory | short article about total movie sales |
| october 25 , 2006 | belgium | 39000 | Nightmares of Glory | article about total movie sales |
| october 25 , 2006 | germany | 135000 | Nightmares of Glory | mid-length article about daily movie sales |
| october 26 , 2006 | austria | 42000 | Nightmares of Glory | article about total movie sales |
| october 26 , 2006 | netherlands | 54000 | Nightmares of Glory | report of national movie sales |
| october 27 , 2006 | united kingdom | 34700 | Nightmares of Glory | article about total movie sales, and interviewing a fictional moviegoer |
| october 26 , 2006 | [not explicitly stated] | 195,000, 60,000 more than yesterday's sales | Nightmares of Glory | article about total movie sales |

**Augmented Table Question:** Rank the movie's sales by country.
**Augmented Table Answer:** Germany (133000+195000), Turkey (146200), Netherlands (54000), Austria (42000), Belgium (39000), United Kingdom (34700)

Figure 5: Demonstration of table edit plan used during the knowledge augmentation component of the MDBench pipeline.

[Knowledge Aggregation] – The ability to align, compare and/or contrast knowledge that may be present. This includes non-numeric knowledge.

**Baseline Example:** Rank the teams by number of wins in the series.

| race | pole position | winning team |
|------|---------------|--------------|
| May 7, 1992 | nico valencia | ferrari |
| May 21, 1992 | mark steedman | bmw |
| June 4, 1992 | bonnie bobcat | mclaren |
| June 18, 1992 | elio muchin | renault |
| July 2, 1992 | tammy tiger | ford |
| July 16, 1992 | tyrell eshar | ferrari |
| July 30, 1992 | alain prost | ferrari |
| August 13, 1992 | tigre trees | renault |

**Answer:** Ferrari, Renault, and T-3 are BMW, McLaren and Ford.
**Answer Rationale:** Ferrari was listed as the winning team three times, Renault twice, and the others once each.
**Commentary:** This is a simple example that required calculating the number of appearances of each team in the 'winning team' column.

**Harder Example:** Identify the top two teams in this race series, and explain any correlation between their success and the weather.

| race | pole position | winning team | notable conditions |
|------|---------------|--------------|--------------------|
| May 7, 1992 | nico valencia | ferrari | sunny + dry |
| May 21, 1992 | mark steedman | bmw | rainy |
| June 4, 1992 | bonnie bobcat | mclaren | heavy rain |
| June 18, 1992 | elio muchin | renault | slick roads |
| July 2, 1992 | tammy tiger | ford | cold and blustery |
| July 16, 1992 | tyrell eshar | ferrari | sunny |
| July 30, 1992 | alain prost | ferrari | overcast |
| August 13, 1992 | tigre trees | renault | damp |

**Answer:** Ferrari finished first and Renault finished second. Ferrari's wins were exclusively in conditions with dry pavement, whereas Renault won only in wet conditions.
**Answer Rationale:** Ferrari had three wins, and Renault had two wins. The rest of the teams had only one. Notably, Ferrari winning races were only in conditions where the roads were presumably dry (sunny+dry, sunny, and overcast), and Renault's wins were only on day where the conditions were wet (slick roads, and damp).
**Commentary:** This answer requires not only understanding the winning teams, but also realizing that there were patterns in the conditions for both teams. Namely, one had to ascertain that Ferrari performed well on dry days, whereas Renault did well on wet roads. This requires aggregating, comparing, and contrasting values across different rows and teams.

Figure 6: Knowledge Aggregation Skill Description

[Multi-hop Reasoning] – The ability to solve problems requiring multiple steps to arrive at the solution.

**Baseline Example:** Which country had the most showings and how many was this in total?

| date | territory | showings |
|---|---|---|
| october 20, 2006 | turkey | 200 |
| october 20, 2006 | belgium | 600 |

**Answer:** Belgium had the most with 600 showings.
**Answer rationale:** Turkey had 200 showings and Belgium had 600. 600 > 200, therefore Turkey had the most showings.
**Commentary:** This is a simple reasoning process as it requires a simple comparison of two values with no additional reasoning required.

**Harder Example:** Which country had the most showings and how many was this in total?

| date | territory | showings |
|---|---|---|
| october 20, 2006 | turkey | 200 |
| october 20, 2006 | belgium | 600 |
| october 25, 2006 | turkey | 500 |

**Answer:** Turkey had the most with 700 showings.
**Answer Rationale:** Turkey had showings on two different days, so the total is 200+500=700 showings. 700 > Belgium's 600, therefore Turkey had the most.
**Commentary:** By adding a new row with complementary information, we necessitate an additional reasoning hop to correctly answer the question. Note that this table was edited specifically such that the answer (Turkey) is flipped from the original answer (Belgium) in the simple example. Edits like these ensure the reasoning cannot be shortcutted (e.g., by simply selecting the row with the highest showings).

Figure 7: Multi-hop Reasoning Skill Description

**[Numeric Reasoning] – The ability to handle numeric values and perform numerical operations**

**Baseline Example:** Rank each day by the total showings.

| date | territory | showings |
|---|---|---|
| october 20, 2006 | turkey | 200 |
| november 21, 2006 | belgium | 600 |
| november 21, 2006 | turkey | 400 |
| november 22, 2006 | belgium | 600 |

**Answer:** November 21st had the most showings with 1000, followed by November 22nd, then October 20th.
**Answer Rationale:** November 21st had 1000 totals showings – 600 in Belgium and 400 in Turkey. This was greater than the 600 on November 22nd and the 200 on October 20th.
**Commentary:** This is a simple case of performing numeric operations, having to sum values over different rows to identify the correct answer.

**Harder Example:** Rank each day by the total sales

| date | territory | showings | Avg. sales per showing ($) |
|---|---|---|---|
| october 20, 2006 | turkey | 200 | 6000 |
| november 21, 2006 | belgium | 600 | 1000 |
| november 21, 2006 | turkey | 400 | 1000 |
| november 22, 2006 | belgium | 600 | 500 |

**Answer:** October 20th had the highest sales, followed by November 21st, then November 22nd
**Answer Rationale:** October 20 had 200 showings * $6000 per showing = $1,200,000. November had 600*$1000 = $600,000 from Belgium and 400*$1000 = $400,000 from Turkey, totalling $1,000,000. November 22 had 600 * $500 = $300,000 in sales.
**Commentary:** This reasoning requires calculating values over two different columns, and then additionally summing values over associated rows (e.g. the november 21 entries).

Figure 8: Numeric Reasoning Skill Description

[Soft Reasoning] – The ability to reason abductively and make informed decision in cases where some uncertainty or fuzziness may be present.

**Simple Example:** Who had the most championships?

| Year | Championship Winner |
|------|---------------------|
| 2008 | Yusef |
| 2009 | Mattingly |
| 2010 | Tigre Trees |
| 2011 | Yusef "Skeeps" Mattingly |
| 2012 | Tigre |
| 2013 | John Smith |
| 2014 | John Smith |
| 2015 | Harrison Chevrolet |

**Answer:** Yusef Mattingly, who had wins in 2008, 2009, and 2011

**Answer Rationale:** Although not clearly stated, some of the entries likely refer to the same person, just sometimes using only the first name, last name, or a nickname. We can reasonably assume 'Yusef', 'Mattingly', and 'Yusef "Skeeps" Mattingly' all refer to the same individual. Similarly, we see both a 'Tigre Trees' and 'Tigre' which likely refer to the same individual.

**Commentary:** This is an example abductive or 'best guess' soft reasoning where one could reasonably assume that some of the entries refer to the same canonical entity/person. Notably, this example is one where a wrong answer would be generated by using a simple exact match heuristic as 'John Smith' appears twice, which is less than Yusef Mattingly.

**Harder Example:** Rank the countries by total sales.

| | Country | Sales ($) | Notes |
|-----------|-----------------------------|---------|-------------------------------|
| October 20 | Turkey | 146200 | |
| October 25 | Belgium | 39000 | |
| October 25 | Germany | 134000 | |
| October 26 | Austria | 42000 | |
| October 26 | Netherlands | 54000 | |
| October 27 | United Kingdom | 534700 | |
| October 26 | <one that was already mentioned> | 195000, roughly 60k more than yesterday's sales. | A follow-up to a prior entry |

**Answer:** United Kingdom, Germany, Turkey, Netherlands, Austria, Belgium

**Answer Rationale:** Most country sales are confined to just one row. However, the final row contains sales information that implicitly refers to a country. We see that this country is already mentioned and that this row is a follow-up to a previous entry with sales numbers. The sales value is $195,000 which is stated as 60k more than the prior day sales. We can use this to ascertain what the country is. Namely, we see that there are two entries for the prior day (October 25). Of these two, Germany's sales were $134,000 which is approximately $60,000 less than $195,000. Belgium's sales were much lower (over $150k less than $195,000). Therefore, we can reasonably conclude that the October 26 entry in mention refers to Germany. Combining the $134,000 from October 25 and $195,000 from October 26, we see Germany's total sales are $329,000, which is less than the United Kingdom, but more than Turkey.

**Commentary:** This problem requires that one notices that the final row can be linked to a prior row. Once this is done, there is some soft reasoning that clearly leads to the proper solution. So, while there is some abduction reasoning required, it is very clear once you put the pieces together.

Figure 9: Soft Reasoning Skill Description

[Temporal Reasoning] – The ability to handle temporal information and dependencies.

**Baseline Example:** How many total showings were there in each month?

| date | territory | showings |
|------|-----------|----------|
| october 20, 2006 | turkey | 200 |
| november 21, 2006 | belgium | 600 |
| november 21, 2006 | turkey | 400 |
| november 22, 2006 | belgium | 600 |

**Answer:** October 2006 had 200 showings, while November had 1,600
**Answer Rationale:** October had just one day with 200 showings. November had 3 showings total, summing to 600+400+600 showings total.
**Commentary:** This is fairly straightforward as we simply sum all rows sharing the same month.

**Harder Example:** How many total showings were there in each month?

| date | territory | showings | notes |
|------|-----------|----------|-------|
| october 20, 2006 | turkey | 200 | Opening day in Turkey |
| november 21, 2006 | belgium | 600 | Opening day in Belgium |
| the week after opening day | turkey | 400 | |
| november 23, 2006 | belgium | 600 | |

**Answer:** October 2006 had 600 showings, while November had 1,200
**Answer Rationale:** In Turkey, the week after opening day fell in the month of October, therefore there were 200 (from opening day) + 400 (from the week after) = 600 showings in October. November had 600+600 = 1,200 showings, all from Belgium.
**Commentary:** We introduce a cross-row dependency here that requires temporal reasoning to solve. Namely, we need to intuit that, given opening day is on October 20th, the week immediately following it must fall within the month of October. Again, we intentionally edit the values in the table (and add a 'notes' column) to ensure that the answer (600 in October, 1200 in November) necessarily required resolving this cross-row dependency.

Figure 10: Temporal Reasoning Skill Description

