# OpenReview forum: "MDBench: A Synthetic Multi-Document Reasoning Benchmark Generated with Knowledge Guidance"
_ICLR.cc/2025/Conference — ICLR 2025 Conference Withdrawn Submission_

### Official Review · Reviewer_wqAP · 2024-11-01

**Soundness:** 2
**Presentation:** 2
**Contribution:** 2
**Rating:** 5
**Confidence:** 4

**Summary:**

This paper presents MDBench, a synthetic dataset for evaluating large language models (LLMs) on multi-document reasoning tasks. MDBench is generated using a process that starts with structured knowledge from tables, which is then augmented with complexities requiring advanced reasoning skills by an LLM. The resulting structured knowledge is then converted into documents, creating a multi-document question-answering (QA) example. The authors claim that this method allows for the controlled generation of challenging examples while mitigating data contamination issues commonly found in real-world datasets. The document then presents experimental results showcasing how MDBench effectively challenges even state-of-the-art LLMs and explores the impact of different reasoning characteristics on model performance.

**Strengths:**

1. The idea of using structured data to generate documents for multi-document reasoning is interesting.
2. The authors tested the benchmark data on a variety of LLMs and provided some analysis based on that.

**Weaknesses:**

1. the dataset is too small to achieve statistical significance - only 300 human verified.
2. the authors didn't clarify their quality control progress in the main text (there are still 2 pages left).
3. examples and demonstration process is difficult to read, making the paper unnecessarily hard to follow.

**Questions:**

In Line 148-150, you mentioned you introduced counterfactual or frictional twists. Could you clarify what are these? Wouldn't these make your datasets less trust-worthy and how do you validate them?

---

### Official Review · Reviewer_cV8a · 2024-11-03

**Soundness:** 1
**Presentation:** 2
**Contribution:** 1
**Rating:** 3
**Confidence:** 4

**Summary:**

The paper presents a new synthetic benchmark intended to test multi-document reasoning abilities. This is generated by starting from an existing knowledge base of relational facts, which through a series of prompts to GPT-4o is first made to have more cross-row dependencies, generates questions and answers based on the rows, then each row is converted into a document, forming a synthetic multi-document benchmark. Various LLMs are tested against this benchmark.

**Strengths:**

- I think that addressing multi-document capabilities of LLMs is a timely question with real-world implications as many tasks are phrased over collections of documents, and there aren't many benchmarks specifically geared to test multi-document performance.
- I appreciate that there’s an explicit description of the desired goals for the benchmark (Section 3.1), and I'm generally on board with the goals.

**Weaknesses:**

- My main concern with this paper is the quality of the dataset, and how much it represents multi-document tasks. Real-world multi-document corpora contain several unique challenges - the context is long, there are many documents, the narrative may be contradicting, complementing, repeating, without a clear sense of order. While one of the stated goals of the dataset is to be "Grounded in Real-World Scenarios", the paper didn't try to prove that MDBench actually contains any of these challenges. To me this is the major thing to prove in the paper yet it wasn't discussed at all. Without it, I cannot trust that the results on MDBench indeed measure or correlate with performance on real-world multi-document tasks.

- Perhaps related, There are no statistical details or demonstrations of the actual documents produced in the dataset, so it's impossible to say if they resemble actual documents. If they are instantiated based on single factoids, I worry that they may actually be rather short? Perhaps containing single sentences or short paragraphs rather than actual document-length texts.

**Questions:**

- How were the documents created? How many words did they contain on average?
- Did the dataset contain real-world challenges? (redundancy, contradiction, etc., see Weaknesses)

- How were the instances validated? The paper only mentions that they were "hand-verify"ed (Page 5). I'd appreciate much more details - who were the validators? What guidelines were they given? What was their agreement? How many instances did they reject and why?

---

### Official Review · Reviewer_d47n · 2024-11-05

**Soundness:** 2
**Presentation:** 2
**Contribution:** 2
**Rating:** 3
**Confidence:** 4

**Summary:**

The authors propose a new benchmark called MDBench. MDBench is created through a synthetic generation process, which operates on condensed structured seed knowledge. The structured knowledge is then converted into a natural text surface form with a document set and corresponding QA example. The authors evaluate the performance of models from several prominent LLM families. They find that MDBench poses a strong challenge.

**Strengths:**

1. This paper introduce a benchmark generation pipeline for multi-documents. The pipeline seems reasonable and useful.

2. The authors focus on diverse multi-document settings in the experiments, such as Multi-hop Reasoning and Numeric Reasoning.

3. The curated benchmark is challenging. Many SOTA LLMs perform unsatisfactorily.

**Weaknesses:**

1. The authors fail to consider other reasoning baseline methods in their benchmark. The authors should explore and evaluate other more related inference baselines in MD settings, such as the related works in Section 2 mentioned in the paper.

2. Also, there are no comparisons with other data synthetic methods, such as the related works in Section 2.

3. The authors have not provided further convincing analysis on why LLMs fail to complete such tasks, or how to improve their performance.

4. The presentation (such as Tables and Figures) needs improving.

**Questions:**

1. Do you try to train a stronger model using your collected datasets?

2. How can you ensure that the synthetic data in the MDBench are correct?

3. Why do you consider one-shot instead of few-shot? Perhaps due to the input length constraint? If that is the case, you can use some long-context methods in LLMs.

4. Can ToT [1] (or other XoT methods) be used in this task like CoT?

[1] Yao, Shunyu, et al. "Tree of thoughts: Deliberate problem solving with large language models." Advances in Neural Information Processing Systems 36 (2024).

---

### Official Review · Reviewer_SLud · 2024-11-06

**Soundness:** 2
**Presentation:** 2
**Contribution:** 3
**Rating:** 5
**Confidence:** 4

**Summary:**

The paper proposes a pipeline for automatically generating a Question Answering (QA) dataset involving multi-document sources, using artificially-augmented, single tables as proxy. The resulting dataset consists of 1k data instances, 300 of which have been manually verified. Evaluation results using popular LLMs with various prompting techniques (incl. zero-shot and one-shot settings with and without Chain-of-Thought) showcase that even some of the frontier models struggle to achieve exact match performance greater than 70%.

**Strengths:**

- Interesting findings comparing different LLMs' capabilities for multi-document-based against table-based reasoning.
- Interesting idea for generating automatically a multi-document QA dataset using single tables as proxy.
- The proposed benchmark showcases the difficulty of frontier models to address challenges associated with the multi-document QA setting, and it may be useful for motivating further work in this space.

**Weaknesses:**

- The paper does not properly relate itself to prior work within the domain of multi-document, synthetic dataset generation. What are the main aspects that differentiate it from Schnitzler et al. (2024) and Sprague et al. (2023)?
- The quality guarantees are not fully convincing due to absence of statistics that correlate the human judgements with the automatically-validated data instances.
  - Details about the human evaluation process for the 300 manually annotated cases are missing from the manuscript. How many annotators were assigned per data instance, and what was the inter-annotator agreement?
- The presented statistics and experiments do not clearly indicate whether the benchmark generation goals (as outlined in Section 3.1) are observed in the resulting dataset.
  - For instance, how many of the questions are fully grounded in real-world concepts? I would assume that some of these questions may be answerable without requiring the input table or document(s) as prerequisites.

Julian Schnitzler, Xanh Ho, Jiahao Huang, Florian Boudin, Saku Sugawara and Akiko Aizawa. 2024. MoreHopQA: More Than Multi-hop Reasoning. Preprint, arXiv:2406.13397.

Zayne Sprague, Xi Ye, Kaj Bostrom, Swarat Chaudhuri and Greg Durrett. 2023. MuSR: Testing the Limits of Chain-of-thought with Multistep Soft Reasoning. Preprint, arXiv:2310.16049.

**Questions:**

- The paper could benefit from a detailed comparison of how LLMs perform with different input context window sizes, especially for multi-document reasoning use cases. I would expect that with longer input documents, the performance benefits of a more compressed, tabular representation would become more apparent.
- Were there any generated instances that were filtered out due to not meeting the quality standards set by the human annotators or the automatic validation method? What is the portion of the filtered-out cases with respect to the size of the resulting dataset?
- I was unable to find an example of the resulting documents in the manuscript. Including an example of a complete data instance in the Appendix would enhance the clarity of the paper.
- Are there any particular considerations for selecting which tables from TabFact should be used?
- Typo: There is an extra "z" at the bottom row of the last column of Table 4.

---

### Note · Authors · 2024-12-16

I have read and agree with the venue's withdrawal policy on behalf of myself and my co-authors.